# Tannic acid, an IL-1β-direct binding compound, ameliorates IL-1β-induced inflammation and cartilage degradation by hindering IL-1β-IL-1R1 interaction

Hae-Ri Lee[1]☯, Young-Jin Jeong[1]☯, Joong-Woon Lee[1], JooYeon Jhun[2,3,4], Hyun Sik Na[2,3,4], Keun-Hyung Cho[2,3,4], Seok Jung Kim[5], Mi-La Cho[2,3,6], Tae-Hwe Heo📷[1]*

1 Laboratory of Pharmacoimmunology, Integrated Research Institute of Pharmaceutical Sciences and BK21 FOUR Team for Advanced Program for Smart Pharma Leaders, College of Pharmacy, The Catholic University of Korea, Seoul, Republic of Korea, 2 The Rheumatism Research Center, Catholic Research Institute of Medical Science, College of Medicine, The Catholic University of Korea, Seoul, Korea, 3 Lab of Translational Immuno Medicine, Catholic Research Institute of Medical Science, College of Medicine, The Catholic University of Korea, Seoul, South Korea, 4 Department of Biomedicine & Health Sciences, College of Medicine, The Catholic University of Korea, Seoul, Republic of Korea, 5 Department of Orthopedic Surgery, Uijeongbu St. Mary's Hospital, College of Medicine, The Catholic University of Korea, Seoul, Republic of Korea, 6 Department of Medical Life Sciences, College of Medicine, The Catholic University of Korea, Seoul, Republic of Korea

☯ These authors contributed equally to this work.
* thhur92@catholic.ac.kr

**Data Availability Statement:** All relevant data are within the paper and its Supporting Information files.

## Abstract

Interleukin-1β (IL-1β) is one of the most potent pro-inflammatory cytokines implicated in a wide range of autoinflammatory, autoimmune, infectious, and degenerative diseases. Therefore, many researchers have focused on developing therapeutic molecules that inhibit IL-1β-IL-1 receptor 1 (IL-1R1) interaction for the treatment of IL-1-related diseases. Among IL-1-related diseases, osteoarthritis (OA), is characterized by progressive cartilage destruction, chondrocyte inflammation, and extracellular matrix (ECM) degradation. Tannic acid (TA) has been proposed to have multiple beneficial effects, including anti-inflammatory, anti-oxidant, and anti-tumor activities. However, it is unclear whether TA plays a role in anti-IL-1β activity by blocking IL-1β-IL-1R1 interaction in OA. In this study, we report the anti-IL-1β activity of TA in the progression of OA in both *in vitro* human OA chondrocytes and *in vivo* rat OA models. Herein, using-ELISA-based screening, natural compound candidates capable of inhibiting the IL-1β-IL-1R1 interaction were identified. Among selected candidates, TA showed hindering IL-1β-IL-1R1 interaction by direct binding to IL-1β using surface plasmon resonance (SPR) assay. In addition, TA inhibited IL-1β bioactivity in HEK-Blue IL-1-dependent reporter cell line. TA also inhibited IL-1β-induced expression of inducible nitric oxide synthase (NOS2), cyclooxygenase-2 (COX-2), IL-6, tumor necrosis factor-alpha (TNF-α), nitric oxide (NO), and prostaglandin E2 (PGE2) in human OA chondrocytes. Moreover, TA downregulated IL-1β-stimulated matrix metalloproteinase (MMP)3, MMP13, ADAM metallopeptidase with thrombospondin type 1 motif (ADAMTS)4, and ADAMTS5, while upregulating collagen type II (COL2A1) and aggrecan (ACAN). Mechanistically, we confirmed that TA suppressed IL-1β-induced MAPK and NF-κB activation. The protective

**Funding:** This work was supported by the Basic Science Research Program through the National Research Foundation of Korea (NRF) funded by the Ministry of Education, Science, and Technology (Grant number: NRF-2018R1A6A1A03025108 and 2021R1A2C2009782). This study was supported by the Research Fund, 2023 of The Catholic University of Korea.

**Competing interests:** The authors have declared that no competing interests exist.

effects of TA were also observed in a monosodium iodoacetamide (MIA)-induced rat OA model by reducing pain and cartilage degradation and inhibiting IL-1β-mediated inflammation. Collectively, our results provide evidence that TA plays a potential role in OA and IL-1β-related diseases by hindering IL-1β-IL-1R1 interaction and suppressing IL-1β bioactivity.

## 1. Introduction

Interleukin-1β (IL-1β) is a potent proinflammatory cytokine capable of inducing various inflammatory responses in various cell types [1, 2]. Binding of IL-1β to the cellular IL-1 receptor permits the association of the IL-1 receptor accessory protein (IL-1RAcP) to initiate a signaling cascade and induce the expression of inflammatory mediators involved in disease progression [3]. IL-1 receptors (IL-1R) are expressed in nearly all cells and tissues; thus, overproduction of IL-1β is a critical pathogenic mediator of various IL-1-related autoinflammatory, autoimmune, infectious, and degenerative diseases. Thus, many researchers have considered inhibiting the interaction between IL-1β and IL-1R1 as a therapeutic target for the treatment of various IL-1-related diseases. For instance, anakinra binds to IL-1R1 and inhibits its interaction with IL-1β. The anti-IL-1β activity of anakinra exhibit reduced mortality via anti-hyperinflammatory efficacy in patients with severe COVID-19 pneumonia [4]. It also showed anti-hyperinflammatory efficacy in other hyperinflammatory situations, including the cytokine release syndrome and immune effector cell-associated neurotoxicity syndrome observed during antitumoral chimeric antigen receptor (CAR) T-cell therapy [5, 6]. In addition, there are several different approaches being tried to target the interaction between IL-1β and IL-1R1, and many studies have reported the therapeutic efficacies of IL-1 blockers in IL-1-related diseases [7, 8].

To date, three drugs targeting the IL-1β signaling pathway have been approved for clinical treatment. Anakinra is a recombinant IL-1 receptor antagonist; rilonacept, a dimeric fusion protein consisting of human IL-1R and IL-1RAcP; and canakinumab, an IL-1β neutralizing antibody [9]. In recent years, a number of studies have verified the efficacy of these approved drugs for various IL-1-related diseases, including autoimmune diseases such as osteoarthritis, gout, juvenile idiopathic arthritis, and diabetes [9, 10]. Therefore, the trend of these research reports is to focus on the possibility of its application as a therapeutic agent for various diseases related to IL-1 through the development of drugs that block the interaction between IL-1β and IL-1R1.

Among IL-1-related diseases, osteoarthritis (OA), the most prevalent form of arthritis, is characterized by progressive cartilage destruction, chondrocyte inflammation, and extracellular matrix (ECM) degradation [11]. Persistent chronic inflammation has a significant impact on the onset and progression of osteoarthritis. Inflammatory mediators, particularly cytokines, contribute to accelerated joint tissue degradation and pain [12]. Since accumulating evidence has demonstrated that IL-1β-mediated inflammation is important in the progression of OA, blocking IL-1β signaling may be used as an additional therapeutic strategy in drugs for OA. Therefore, it would be a good strategy to use IL-1β-mediated OA progression as an *in vivo* research model to evaluate the therapeutic efficacy of compounds with anti-IL-1β activity.

Tannins are naturally occurring plant phenolic compounds that are classified into hydrolyzed and condensed tannins. Among them, tannic acid (TA), a hydrolyzed tannin, is found not only in drinks such as black tea, coffee, and red wine, but also in fruits and various vegetables [13]. TA exhibits multiple beneficial effects including anti-inflammatory, anti-oxidant,

anti-fungal, and anti-tumor activities [14–17]. In addition, TA also inhibits NLRP3 inflamma-some-mediated IL-1β production via blocking NF-κB activation in macrophages [18]. How-ever, it is unclear whether TA shows the anti-IL-1β activity by blocking the interaction between IL-1β and IL-1R1.

Therefore, in this study, we aimed to unveil the anti-IL-1β property of TA, which blocks the interaction between IL-1β and IL-1R1, and to determine the applicability of TA as an anti-IL-1β therapeutic agent using human OA chondrocytes and a rat OA model.

## 2. Materials and methods

### 2.1. Screening the library to discover IL-1β-blocking candidates

ELISA-based binding assays were performed for IL-1β-blocking candidates to block the inter-action between IL-1β and IL-1R1 using 2303 natural compound libraries (TargetMol L-6000, Wellesley Hills, MA, USA; Selleckhem L-1400, Matsonford Road Radnor, PA, USA). In brief, ELISA plates were coated with 100 ng/well of recombinant human IL-1β (R&D Systems, Min-neapolis, MN, USA) in PBS (Sigma-Aldrich, Burlington, MA, USA) at 4˚C overnight. After coating, the plates were washed with PBST [PBS containing 0.05% Tween-20 (Sigma)] and blocked with PBSA [PBS containing 1% BSA (Bovogen Biologicals, East Keilor, Australia)] for 1 h at room temperature (RT). After an additional wash with PBST, each natural compound was diluted to 20 μM in PBSAT (PBS containing 1% BSA and 0.05% Tween-20), added to each well, and incubated for 2 h at RT. After three washes with PBST, recombinant human IL-1R1 protein (125 ng/ml, Abcam, Cambridge, UK) was added and incubated for 2 h. Subsequently, horseradish peroxidase (HRP)-conjugated anti-human IgG Fc (1:2000, Bethyl Laboratories, Montgomery, TX, USA) was added and incubated for 1 h. After washing, the TMB solution (Surmodics, Inc., Eden Prairie, MN, USA) was added to each well and incubated. Optical den-sity (OD) at 450 nm was determined using an Epoch microplate spectrophotometer (BioTek Instruments, Santa Clara, CA, USA). Data are presented as the mean of triplicate samples. Dose-dependency tests of selected natural compounds for blocking recombinant human IL-1β and IL-1R1 interactions. Every step was identical to the primary screening except for the con-centrations of the selected natural compounds (10, 40, or 160 μM) and washing buffer (PBS containing 0.05% Tween-20 and 0.01% Triton X-100). Binding of IL-1β and IL-1R1 without any single compound was used as a positive control (P.C.), and nonspecific binding of IL-1β and HRP-conjugated anti-human IgG secondary antibody without any single compound and IL-1R1 was used as a negative control (N.C.).

### 2.2. Surface plasmon resonance (SPR) analysis

Surface plasmon resonance (SPR) analysis was performed using a Biacore T200 instrument (GE Healthcare, Arlington Heights, IL, USA) with a CM5 sensor chip (GE Healthcare) at 25˚C. To test TA (Sigma-Aldrich) binding of IL-1β, recombinant human IL-1β was immobi-lized on a CM5 sensor chip using the amine-coupling method, according to standard proto-cols. Various concentrations of TA were injected into the flow system at a flow rate of 20 μL/min for 300 s and allowed to dissociate for 600 s. The $K_D$ values of TA against IL-1β protein were obtained using the T200 BIA evaluation software (GE Healthcare).

### 2.3. Neutralizing bioassay with IL-1β-dependent HEK-Blue IL-1β cell

To examine IL-1β neutralizing activity of TA, HEK-Blue IL-1β cells (InvivoGen, San Diego, CA, USA) were used. Cells were cultured in DMEM (WELGENE, Gyeongsangbuk-do, Korea) containing 10% fetal bovine serum (FBS, Corning, Corning, NY, USA), 100 units/ml

penicillin, 100 mg/ml streptomycin (Invitrogen), 100 μg/ml Normocin (InvivoGen), and 100 μg/ml Zeocin (InvivoGen), and cultured at 37˚C in 5% $CO_2$ humidified incubator. Cells ($5\times10^4$ cells/well) were seeded in a 96-well plate and stimulated with human recombinant IL-1β (10 ng/ml) with or without various concentrations of TA (0.78, 1.56, 3.125, 6.25, 12.5, 25, 50, or 100 μM). After a 24 h incubation, SEAP activity was detected using QUANTI-Blue™ (Invitrogen) and the OD at 620 nm. Cell viability was measured by analyzing the OD at 450 nm using D-Plus CCK (DONGIN LS, Gyeonggi-do, Korea).

## 2.4. Human OA chondrocytes and stimulation

All protocols were approved by the institutional review board of Seoul St. Mary's Hospital (KC21SISI0337) and were performed in accordance with the Declaration of Helsinki. Human articular cartilage was acquired from patients for replacement arthroplasty or joint replacement surgery, and the chondrocytes were obtained from the cartilage and maintained in DMEM (WELGENE) containing 10% FBS (Corning), 100 units/mL of penicillin, and 100 mg/mL of streptomycin (Invitrogen), and cultured at 37˚C in 5% $CO_2$ humidified incubator. Human OA chondrocytes were seeded at $3\times10^5$ cells/well or $2\times10^5$ cells/well into 6-well or 12-well plates, respectively. Following 24 h of starvation with insulin-transferrin-selenium (ITS-G, Thermo Fisher, Waltham, MA, USA), cells were stimulated with 10 ng/ml IL-1β with or without various concentrations of TA (0.5, 1, or 2 μM), recombinant human IL-1R1 Fc (1 μg/ml, Abcam), or anti-IL-1β neutralizing antibody (1 μg/ml, Abcam) in serum-free DMEM for 1, 3, or 48 h. Recombinant human IL-1R1 Fc or anti-IL-1β neutralizing antibody were used as a positive control and medium only group was used as a negative control. The supernatant and cells were collected for further analysis.

## 2.5. Enzyme-linked immunosorbent assay (ELISA)

The concentrations of IL-6, TNF, and PGE2 in the cell supernatants were measured using commercial human IL-6, TNF (R&D Systems), and PGE2 ELISA kits (R&D Systems), following the manufacturer's protocol. As an indicator of NO production, nitrite was measured in the culture medium using the Griess Reagent System (Promega, Madison, WI, USA), following the manufacturer's protocol. Optical density at 450 nm was determined using an Epoch microplate spectrophotometer (BioTek Instruments).

## 2.6. Western blot analysis

Total lysates of human OA chondrocytes were prepared using RIPA buffer (Biosesang, Gyeonggi-do, Korea). Primary antibodies against MMP3, MMP13 (1:1000 dilution, Cell Signaling Technology, Danvers, MA, USA), COL2A1 (1:1000 dilution, Abcam), ACAN (1:500 dilution, Invitrogen), p38, p-p38, ERK, p-ERK, JNK, p-JNK, p65, p-p65, IκBα, p-IκBα, or β-actin (1:1000 dilution, Cell Signaling Technology) and appropriate HRP-conjugated secondary antibodies (1:2000 dilution, Bethyl) were used. Enhanced chemiluminescence (ECL) detection kit (Thermo Fisher Scientific) was used to detect for specific bands. Chemiluminescent signals were analyzed on a ChemiDoc XRS gel imaging system (Bio-Rad Laboratories, Hercules, CA, USA), and intensity of specific bands was quantified using Image J software (NIH, MD, USA).

## 2.7. Reverse transcription–quantitative polymerase chain reaction (RT-qPCR)

Total RNA was extracted from chondrocytes using TRIzol® reagent (Ambion, Carlsbad, CA, USA), and 1 μg of total RNA was used for cDNA synthesis using a reverse transcription

reagent kit (Meridian Bioscience, Cincinnati, OH, USA), according to the manufacturer's protocol. RT-qPCR was performed using the SYBR® Green real-time PCR kit (Mbiotech, Seoul, Korea) on a Bio-Rad Real-Time PCR system (Bio-Rad Laboratories). Primers used for RT-qPCR reactions are listed in S1 Table. The relative expression levels of each gene were normalized to the internal reference gene *GAPDH*.

### 2.8. Animals

Male Wistar rats (Six-week-old, 230–280 g) were purchased from Central Lab Animal Inc. (Seoul, Korea). The animal experiments in this study were performed followed ethical guidelines for animal use and approved (Permit Number:2019-0302-01) by the Laboratory Animals Welfare Act (Korea), the Guide for the Care and Use of Laboratory Animals, and the Guidelines and Policies for Rodent Experiments of the Institutional Animal Care and Use Committee (IACUC) of the School of Medicine, Catholic University of Korea.

### 2.9. Monosodium iodoacetate model of OA and treatment with TA

Osteoarthritis rat model was induced by a single intra-articular injection of monosodium iodoacetate (MIA; Sigma-Aldrich). After anesthetization, the rats were injected with MIA (1 mg) into the intra-articular space of the right knee. Three days after MIA injection, the rats were received daily oral administration of TA (50 mg/kg) or celecoxib (50 mg/kg, positive control) during 21 days and were sacrificed using isoflurane anesthesia for further analysis.

### 2.10. Assessment of pain and weight-bearing measurement

Nociception tests in MIA-induced OA rats were performed by placing the rats on a metal mesh in the chamber and using a dynamic plantar sensory meter (Ugo Basile, Gemonio, Italy). When a microscopic plastic monofilament touched the paw and the monofilament's force was gradually increased and stimulated, it caused pain in the rat, which was indicated by the rat's paw withdrawal. The force used to produce the foot withdrawal reflex was measured. Rats acclimatized in an acrylic holder for a certain period of time fixed their feet on the pad and measured the weight balance of both feet for 5 seconds. The weights on the unguided and guided legs were determined. The percentage weight balance was obtained by comparing legs with and without OA.

### 2.11. Histopathological analyses

Three weeks after MIA injection, joint tissues were sectioned at 5 μm thickness and stained with safranin O and hematoxylin and eosin (H&E). Stained tissue samples were analyzed using the Osteoarthritis Research Society International (OARSI) and Mankin scoring systems [19].

### 2.12. Immunohistochemistry

Paraffin-embedded samples were incubated with the specific primary antibodies against IL-1β (1:500 dilution, Abcam, UK), IL-6 (1:500 dilution, Novus Biologicals, Centennial, CO, USA), and MCP-1 (1:500 dilution, Abcam). The tissue samples were then incubated with HRP-conjugated secondary antibody for 30 min. Visualization was conducted using the 3, 3-diaminobenzidine chromogen (Dako, Carpinteria, CA, USA). The percentage of positive cells was analyzed with HDAB (Hematoxylin & DBA) using the image J program (NIH, MD, USA).

### 2.13. Statistical analyses

Statistical analyses were performed using the nonparametric Mann–Whitney U-test for comparisons between two groups and one-way ANOVA with the Bonferroni post-hoc test for multiple comparisons. All graphs were constructed using GraphPad Prism, version 6 (GraphPad Software, San Diego, CA, USA). Data are presented as the mean ± standard deviation (SD). For all comparisons, *p*-value of $< 0.05$ was considered statistically significant.

## 3. Results

### 3.1. Screening and affinity determination of IL-1β-blocking candidate

Natural compound libraries were screened using an ELISA-based binding assay to discover candidate compounds that block IL-1β–IL-1R1 interaction. In the primary screening, TA inhibited the IL-1β-IL-1R1 interaction to 58.88 ± 0.89% compared to the control group of IL-1β-IL-1R1 interaction without any natural compounds (Fig 1A, orange bar; TA). Secondary screening was performed to confirm whether TA could inhibit the interaction between IL-1β and IL-1R1 under harsher conditions. Thus, every step of the ELISA-based binding assay was identical to the primary screening except for the concentrations of TA (10, 40, or 160 μM) and washing buffer (PBS containing 0.05% Tween-20 and 0.01% Triton X-100). As shown in Fig 1B, the interaction between IL-1β and IL-1R1 was inhibited by TA in a dose-dependent manner under harsh conditions. To verify the anti-IL-1β bioactivity of TA, we used the HEK-Blue IL-1β reporter cell line, which was engineered to detect IL-1β-induced activation of NF-κB and AP-1 through a secreted embryonic alkaline phosphatase (SEAP) reporter gene. TA reduced IL-1β-induced SEAP secretion (Fig 1C left panel, $IC_{50} = 13.31$ μM) and anti-IL-1β activity of TA showed a dose-dependent manner (Fig 1C, right panel). In addition, the cytotoxic effect of TA did not show a significant difference up to 100 μM compared to the medium alone group (Fig 1D). The ELISA-based binding assay and HEK-Blue IL-1β cell bioassay showed that TA blocked the IL-1β-IL-1R1 interaction and IL-1β-induced bioactivity. Therefore, an SPR assay was performed to investigate whether TA could directly bind to IL-1β. We first examined the kinetics of the interaction and binding affinity between TA and IL-1β. As shown in Fig 1E, TA bound to IL-1β in a dose-dependent and saturable manner, and the equilibrium dissociation constant $K_D$ was $8.224 \times 10^{-7}$ M. Taken together, it can be inferred that TA is capable of binding to IL-1β to block IL-1β-induced bioactivity by inhibiting IL-1β-IL-1R1 interaction.

### 3.2. Effect of TA on NO, PGE2, IL-6, and TNF production in IL-1β-stimulated human OA chondrocytes

To confirm the inhibitory effects of TA on nitric oxide (NO), PGE2, IL-6, and TNF in IL-1β-stimulated human OA chondrocytes were evaluated using RT-qPCR, Griess reagent, and ELISA kits. As shown in Fig 2A, the mRNA expression levels of *NOS2*, *COX-2*, *IL-6*, and *TNF* increased after IL-1β treatment compared to the control, while IL-1β-induced *NOS2*, *COX-2*, *IL-6*, and *TNF* expression was significantly decreased by TA compared to the IL-1β treatment group. Similarly, the levels of NO, PGE2, IL-6, and TNF production in the culture supernatants were also clearly increased by IL-1β induction; however, TA also significantly reduced NO, PGE2, IL-6, and TNF release in response to IL-1β (Fig 2B). These results indicate that TA inhibited IL-1β-induced inflammatory mediators in human OA chondrocytes.

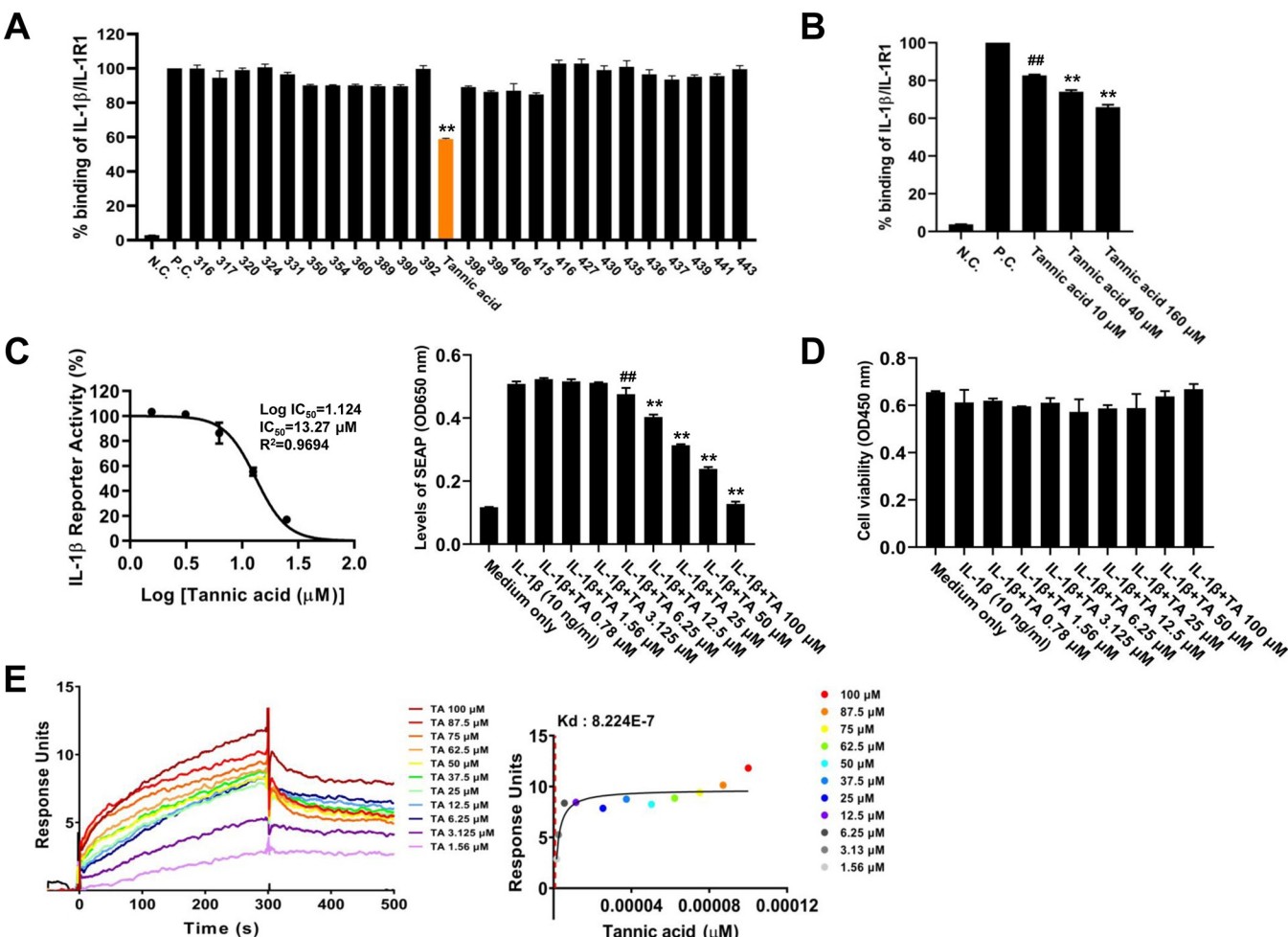

**Fig 1. Screening and affinity determination of small molecule inhibitor of IL-1β.** (A) 2303 natural compound libraries were tested for their ability to inhibit the interaction between recombinant human IL-1β and IL1 Receptor 1. Microtiter 96-well plates were coated with hIL-1β (100 ng/well) overnight and the blocking buffer without any single compound was used as a positive control. Diluted single compounds (20 μM) were added to each well and incubated for 2 h. After washing, recombinant human IL-1R1 (125 ng/ml) was added and incubated for 2 h. Subsequently, HRP-conjugated Anti-Human IgG Fc (1:2000) was added and incubated for 1 h. An OD450 was obtained following the TMB reaction. Data indicate mean ± SD (n = 3). (B) Dose-dependency test of selected natural compound for blocking the interaction between recombinant human IL-1β and IL-1R1. Every step was identical to primary screening except for concentrations of selected natural compound (10, 40, or 160 μM) and washing buffer (PBS containing 0.05% Tween-20 and 0.01% Triton X-100). * $p < 0.05$ compared to the control group of IL-1β and IL1 Receptor 1 interaction without any natural compounds. (C) IL-1β-dependent HEK-Blue IL-1β cells ($5 \times 10^4$ cell/well) were seeded onto a 96-well plate and treated with pre-incubation (20 min) of human IL-1β (10 ng/ml) with various concentrations of TA (0.78, 1.56, 3.125, 6.25, 12.5, 25, 50, or 100 μM). After a 24 h incubation, SEAP activity was assessed using QUANTI-Blue™ and the optical ensity (OD) at 620 nm. Cell viability was measured by analyzing the OD at 450 nm using D-Plus CCK. The IC50 value of tannic acid was determined using the GraphPadPrism 10 software. Data indicate mean ± SD (n = 3). (D) Surface plasmon resonance (SPR) assay was used to analyze the direct binding of tannic acid to human IL-1β. Human IL-1β protein (50 μg/ml) was immobilized on a CM5 sensor chip and various concentrations of TA (1.56, 3.125, 6.25, 12.5, 25, 37.5, 50, 62.5, 75, 87.5, or 100 μM) were injected into the flow system with a flow rate 20 μl/min for 300 s and allowed to dissociate for 600 s. The $K_D$ values of the tannic acid against human IL-1β were obtained using the T200 BIA evaluation software.

## 3.3. Effect of TA on MMPs, ADAMTSs, COL2A1, and ACAN expression in IL-1β-stimulated human OA chondrocytes

We further investigated the effects of TA on IL-1β-stimulated MMP3, MMP13, ADAMTS4, ADAMTS5, COL2A1, and ACAN expression in human OA chondrocytes. The mRNA expression levels of *MMP3*, *MMP13*, *ADAMTS4*, and *ADAMTS5* were clearly increased by IL-1β induction compared with the untreated control. Conversely, TA significantly suppressed IL-

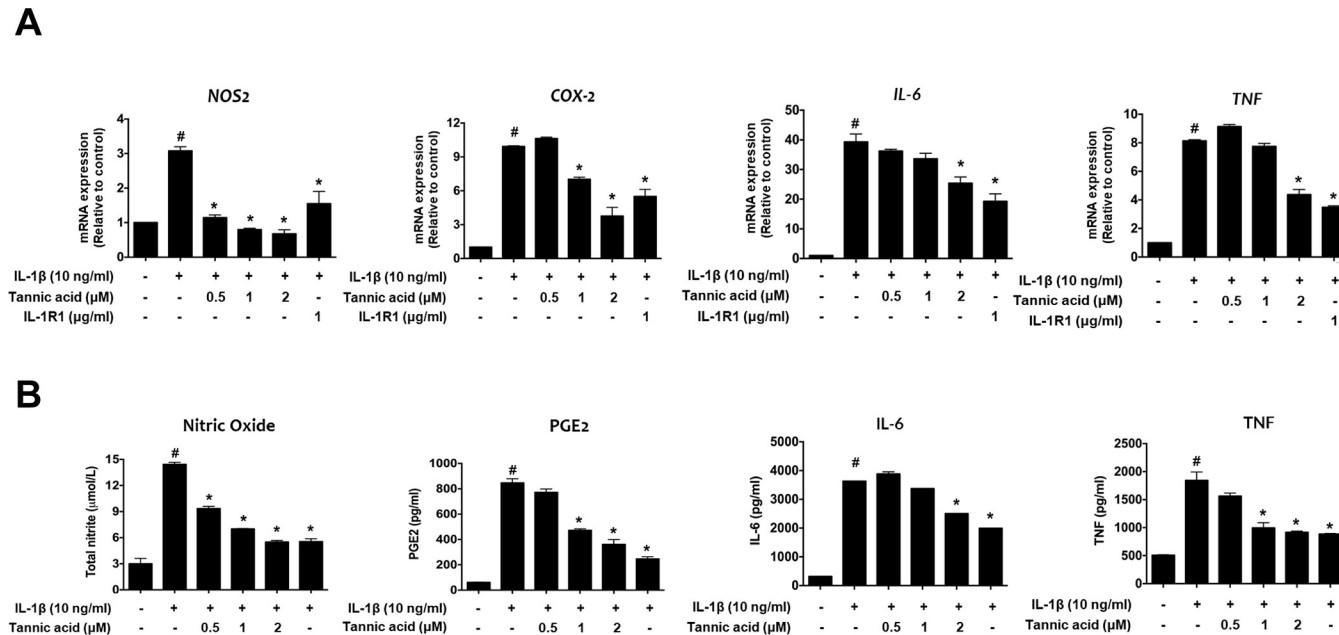

**Fig 2. Effects of TA on the levels of inflammatory mediators in IL-1β-stimulated human OA chondrocytes.** Human articular chondrocytes from OA patients were seeded onto 12-well plates ($2×10^5$ cells/well) and serum-starved cells were co-treated with various concentrations of TA (0.5, 1, or 2 μM) or IL-1R1 (1 μg/ml) and IL-1β (10 ng/ml) for 48 h. (A) The mRNA expression levels of *iNOS*, *COX-2*, *IL-6*, and *TNF* were measured using qRT-PCR. The relative quantity of each gene expression was normalized to the relative quantity of human GADPH. (B) Griess reaction was used to measure the NO levels in the culture supernatants and PGE2, IL-6, and TNF levels in the culture supernatants were evaluated using ELISA. # $p < 0.05$ compared with medium only control group and * $p < 0.05$ compared with IL-1β-treated group.

1β-induced *MMP3*, *MMP13*, *ADAMTS4*, and *ADAMTS5* expression compared to the IL-1β treatment group (Fig 3A). In addition, the mRNA expression of *COL2A1* and *ACAN* was downregulated by IL-1β induction compared to that in the untreated control. In contrast, the downregulated expression of *COL2A1* and *ACAN* induced by IL-1β was increased by TA treatment (Fig 3A). Consistent with the RT-qPCR analysis, western blot analysis showed that the expression of MMP3 and MMP13 increased in response to IL-1β compared to the untreated control, and TA remarkably inhibited their expression (Fig 3B). Additionally, the protein expression of COL2A1 and ACAN was significantly decreased in response to IL-1β, and the application of TA attenuated the IL-1β-induced degradation of COL2A1 and ACAN. Collectively, these results suggest that TA could exert chondroprotective effects in the cartilage via the suppression of IL-1β-induced cartilage-degrading enzymes in human OA chondrocytes.

### 3.4. Effect of TA on the MAPK and NF-κB signaling pathways in IL-1β-stimulated human OA chondrocytes

MAPKs and NF-κB regulate IL-1β-induced inflammatory activity during OA progression. Therefore, we evaluated whether the suppressive effect of TA on IL-1β-induced catabolic factors is regulated by MAPKs and NF-κB in human OA chondrocytes. As shown in Fig 4A, IL-1β significantly enhanced the phosphorylation of p38, ERK, and JNK compared with the untreated control. In contrast, TA showed a remarkable inhibitory effect on the IL-1β-induced phosphorylation of p38, ERK, and JNK. In addition, human OA chondrocytes stimulated with IL-1β induced the phosphorylation of p65 and IκBα, and treatment with TA markedly inhibited the IL-1β-induced phosphorylation of p65 and IκBα (Fig 4B). These results suggest that

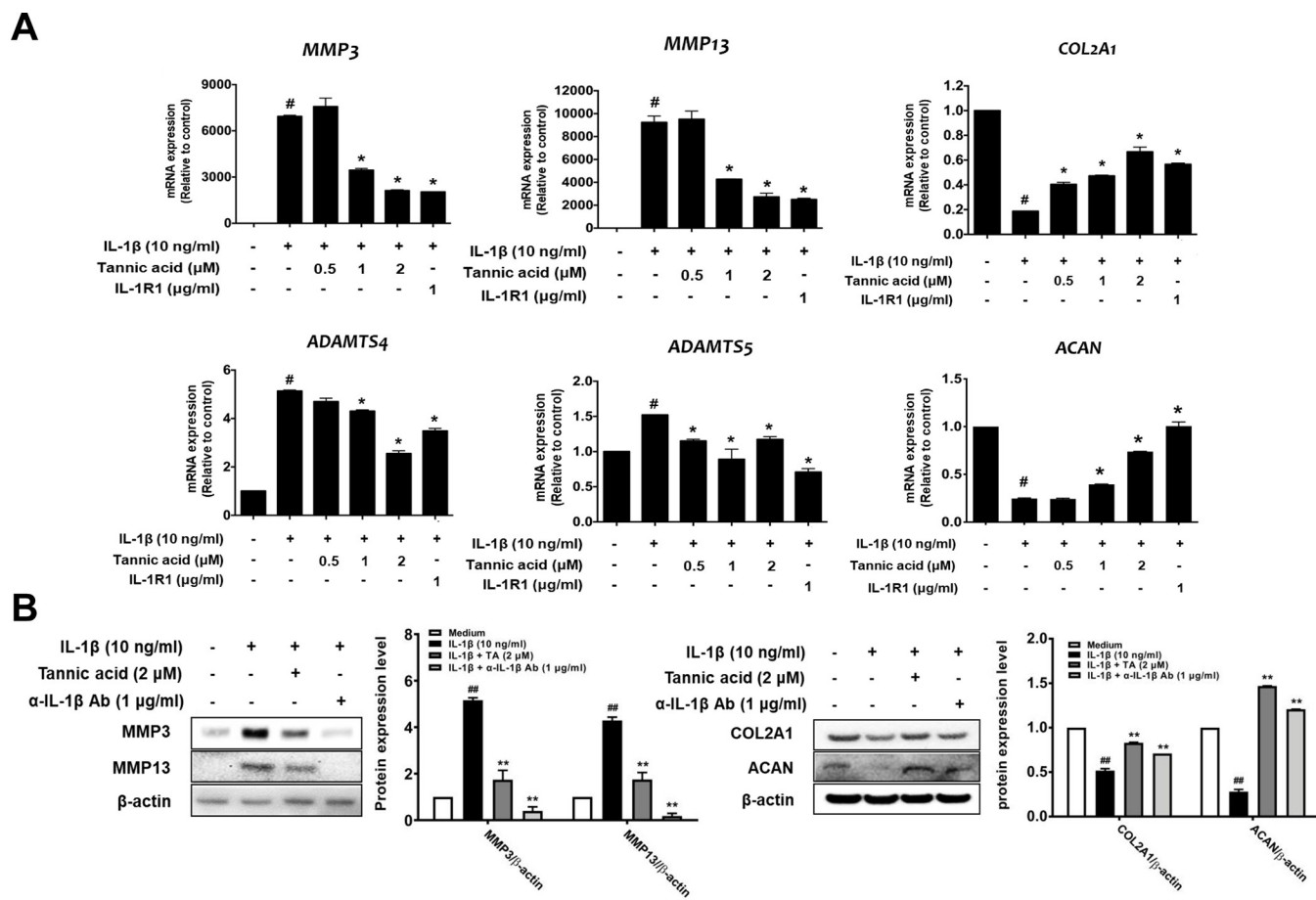

**Fig 3. Effects of TA on the expression of MMPs, ADAMTSs, collagen type II and aggrecan in IL-1β-stimulated human OA chondrocytes.** Human articular chondrocytes from OA patients were seeded onto 6-well plates ($3 \times 10^5$ cells/well) and serum-starved cells were co-treated with various concentrations of TA (2 μM), IL-1R1 (1 μg/ml), or anti-IL-1 neutralizing antibody (1 μg/ml) and IL-1β (10 ng/ml) for 48 h. mRNA and protein expressions of MMPs, ADAMTSs, collagen type II, and aggrecan were detected by qRT-PCR (A) and Western blot analysis (B). The relative quantity of each gene expression was normalized to the relative quantity of human GADPH and β-actin was used as a loading control. # $p < 0.05$ compared with medium only control group and * $p < 0.05$ compared with IL-1β-treated group.

the inhibitory effect of TA on IL-1β-induced MAPK and NF-κB activation is related to its chondroprotective effect of TA on IL-1β-stimulated human OA chondrocytes.

### 3.5. TA suppresses pain and cartilage destruction in MIA-induced OA rat model

To explore the anti-IL-1β efficacy of TA in *in vivo* animal experiments, we used a rat model of one of the IL-1β-related diseases, OA. Weight bearing, paw withdrawal threshold (PWT), and paw withdrawal latency (PWL) were examined to investigate whether TA can reduce pain in the MIA-induced OA rat model. After MIA injection, we observed that the PWT and PWL were significantly increased by TA compared to the vehicle (Fig 5A). These results indicate that oral administration of TA reduces pain. In addition, TA also showed a slight improvement in weight balance, similar to the celecoxib-treated positive control (Fig 5A). These results indicate that TA improved weight bearing, PWT, and PWL in rats with MIA-induced OA. Then, we collected the knee joints and used H&E and safranin O staining to evaluate the degree of cartilage destruction. As shown in Fig 5B, both the total Mankin score and the OARSI score in

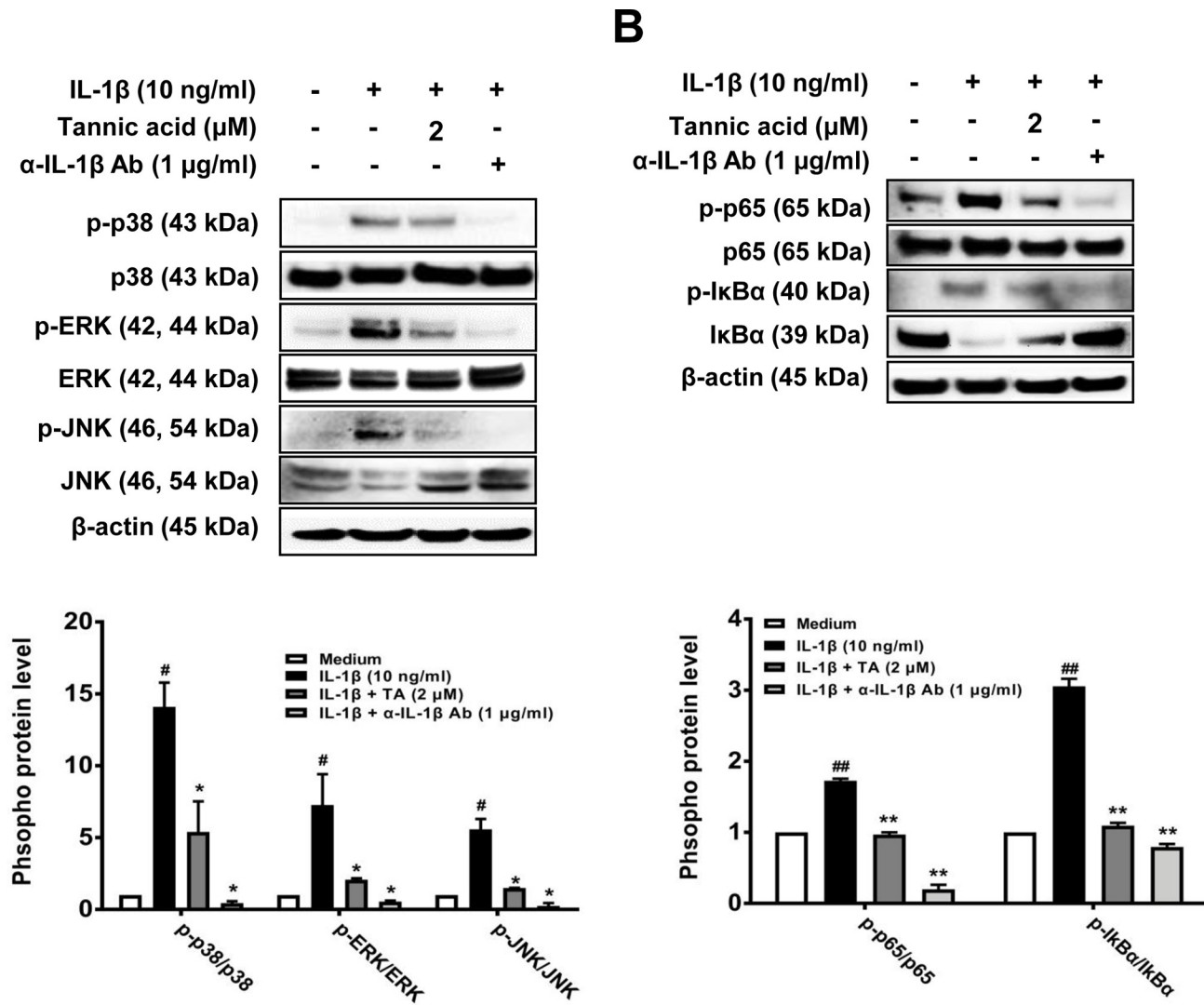

**Fig 4. Effects of TA on IL-1β-induced MAPK and NF-κB activation in human OA chondrocytes.** Human articular chondrocytes from OA patients were seeded onto 6-well plates ($3\times10^5$ cells/well), and serum-starved cells were co-treated with various concentrations of TA (2 μM), or anti-IL-1 neutralizing antibody (1 μg/ml) and IL-1β (10 ng/ml) for 1 h (A) or 3 h (B). Protein expressions of phosphorylated and non-phosphorylated forms of p38, ERK, JNK, p65, and IκBα were detected by Western blot analysis. β-actin was used as a loading control. Band intensity was analyzed using Image J program. # $p < 0.01$ compared with medium only control group and * $p < 0.01$ compared with IL-1β-treated group.

the vehicle group were markedly higher than those in the TA-treated group. In addition, TA significantly reduced articular cartilage damage and proteoglycan destruction compared with vehicle treatment in MIA-induced OA rats. Taken together, these data indicate that TA can inhibit the development of OA in MIA-induced OA rats. To further investigate the effect of TA on the expression of inflammatory mediators, IL-1β, IL-6, and MCP-1, in cartilage from the MIA-induced OA rat model was evaluated via immunohistochemistry. As shown in Fig 5B, IL-1β, IL-6, and MCP-1 were highly expressed in the vehicle group. In contrast, the expression levels of IL-1β and IL-6 were significantly decreased by TA, while the expression level of MCP-1 showed a decreasing trend in the TA-treated group. Therefore, the current results demonstrate that TA exerts a protective effect against OA development by inhibiting cartilage destruction and synovial inflammation in MIA-induced OA rats.

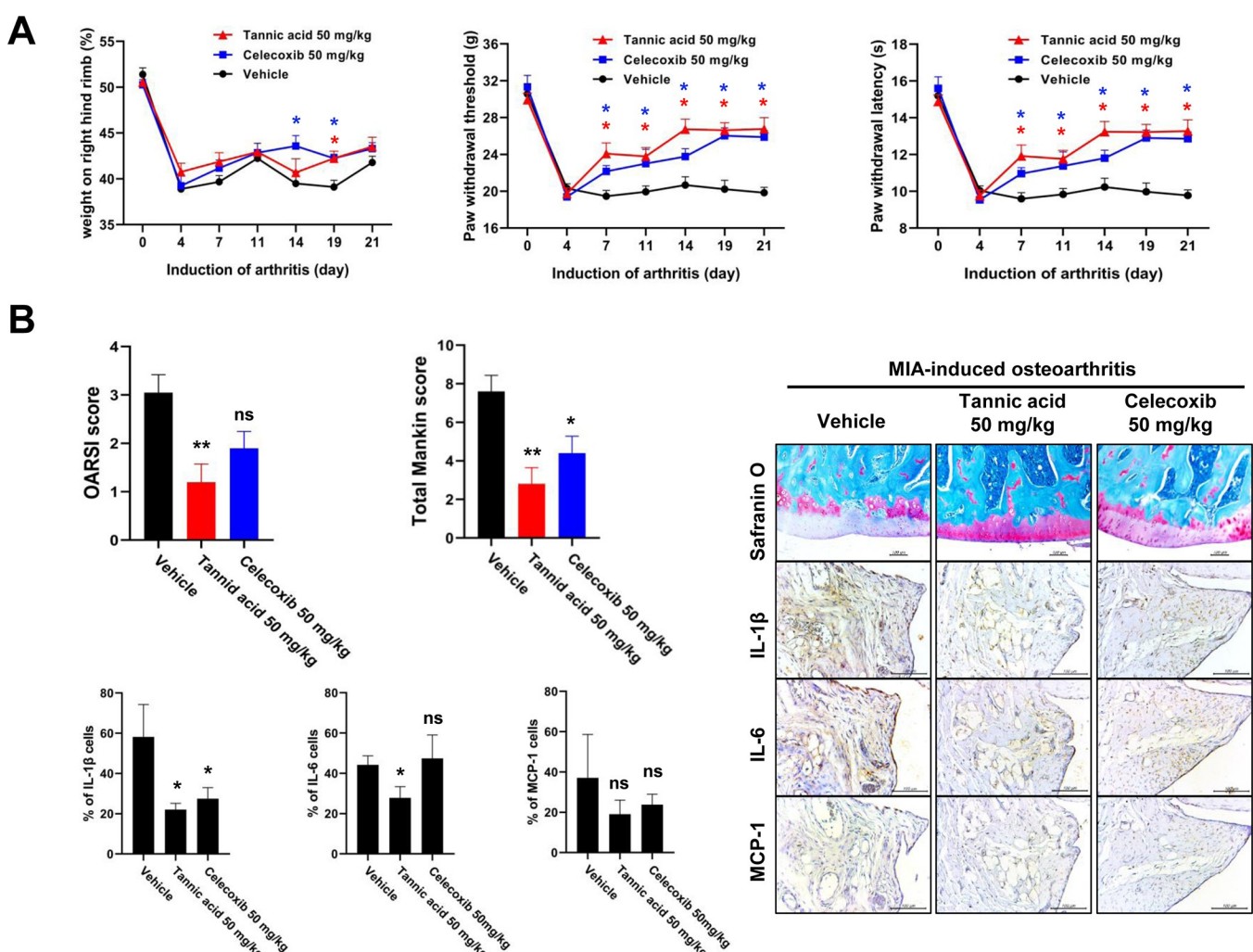

**Fig 5. Therapeutic effect of TA in MIA-induced OA rat model.** (A) Rats were injected with 1 mg of MIA in the right knee. TA (50 mg/kg) was administered orally daily for 3 days after MIA injection. Behavioral tests of mechanical hyperalgesia were evaluated using a dynamic plantar esthesiometer (n = 5). Paw withdrawal latency (PWL) and paw withdrawal threshold (PWT) were conducted right before the administration of TA. The weight balance of MIA-treated rats was analyzed using a capacitance meter. Celecoxib (50 mg/kg) was used as a positive control. Data are presented as the mean ± SD of three independent experiments. * $p < 0.05$ compared with vehicle group. (B) The knee joints from the OA rats treated with either TA, celecoxib, or vehicle control were stained with HE and Safranin O. The joint lesions were graded on a scale of 0–13 using the modified Mankin scoring system, giving a combined score for cartilage structure, cellular abnormalities, and matrix staining. Immunohistochemical analysis was performed to identify the expression of IL-1β, IL-6, and MCP-1 in the articular cartilage. The data are expressed as mean ± SEM for five animals per group. * $p < 0.05$, ** $p < 0.01$, and *** $p < 0.001$ compared with the MIA-induced OA group.

## 4. Discussion

IL-1β is an important regulator in inflammatory immune responses and plays an essential role in a wide range of diseases, such as cancer, rheumatoid arthritis, and atherosclerosis [20, 21]. Therefore, a method of inhibiting IL-1β signaling by blocking the IL-1β:IL-1R1 interaction is considered a target for the development of therapeutic agents for IL-1β-related diseases [22–24]. Currently, there are three FDA-approved blockers that inhibit interleukin-1β signaling and are used safely in the clinic. These drugs include IL-1R antagonist (anakinra), IL-1 soluble receptor (rilonacept), and neutralizing IL-1β antibody (canakinumab). Despite the efficient use of these drugs in the clinic, all of them are biologics; thus, they have the potential for resistance, requirement of

parenteral administration, and high treatment cost; hence, many studies have focused on developing small-molecule inhibitors [9, 25]. Therefore, the aim of present study was to identify small-molecule inhibitors that target IL-1β and to determine their anti-IL-1β efficacy by inhibiting the interaction between IL-1β and IL-1R1. In addition, its applicability as an anti-IL-1β therapeutic agent should be confirmed using an *in vivo* IL-1β related disease animal model.

Here, we conducted ELISA-based screening to identify natural compounds that could inhibit the interaction between IL-1β and IL-1R1. Among the 2303 compounds, we found that TA could directly bind to IL-1β and inhibit the IL-1β-induced activation of NF-κB and AP-1 in HEK-Blue IL-1β reporter cells. TA showed an anti-IL-1β bioactivity against expression of catabolic and anabolic factors in IL-1β-stimulated human OA chondrocytes via inhibition of IL-1β-induced MAPK and NF-κB activation. In addition, TA exerted anti-IL-1β activity in a MIA-induced rat OA model, and the protective effects of TA were confirmed by measuring pain control, cartilage destruction, and IL-1β-induced inflammatory responses. These results indicate that TA attenuated IL-1β-mediated inflammation by hindering the IL-1β:IL-1R1 interaction to diminish signaling downstream of IL-1β in *in vitro* and *in vivo* OA models.

TA is a natural hydrolyzable polyphenol that has been reported to have various efficacies in various research fields of biomedical for many years owing to its unique biochemical properties. In addition, a growing body of research about TA have demonstrated that TA has several therapeutic effects, such as antibacterial, anti-oxidant, antiviral, anti-inflammation, and anticancer activities [26]. In particular, it has been also reported that TA has the effect of inhibiting NLRP3 inflammasome-mediated IL-1β production by blocking NF-κB activation in macrophages [18]. However, no studies have shown that TA directly inhibits IL-1β–IL-1R1 interaction. In this study, we found that TA reduced IL-1β-induced SEAP secretion in a dose-dependent manner in a HEK-Blue IL-1β reporter cell line (Fig 1C). Moreover, we observed that TA showed binding activity to IL-1β in a dose-dependent and saturable manner (Fig 1D). Thus, we inferred that TA is capable of binding to IL-1β to block IL-1β-induced bioactivity by inhibiting the IL-1β–IL-1R1 interaction. Most studies on IL-1β signaling inhibition have reported that inhibition of IL-1β signaling has therapeutic effects in various inflammatory diseases such as rheumatoid arthritis, osteoarthritis, and systemic inflammatory conditions [7, 12, 27]. Further experiments were performed to determine whether TA, which directly binds to IL-1β and inhibits IL-1β signal transduction, exhibits therapeutic efficacy in IL-1β-related diseases.

Chondrocytes and the extracellular matrix are essential for the structure and function of articular cartilage, and IL-1β stimulates the production of various inflammatory mediators, including cytokines, NO, and PGE2 [28, 29]. IL-1β also stimulates chondrocytes to induce the secretion of MMPs and ADAMTSs, which degrade the extracellular matrix, such as COL2A1 and ACAN [30, 31]. Thus, upon stimulation of chondrocytes by IL-1β, the expression of factors related to OA pathology is regulated through NF-kB and MAPK signaling, and thus, NF-kB and MAPKs are major pathways involved in OA disease progression [32, 33]. Here, we found that TA reduced IL-1β-induced expression of inflammatory mediators, MMPs, and ADAMTSs, thereby inhibiting the degradation of COL2A1 and ACAN (Figs 2 and 3). It was also observed that the anti-catabolic and anti-inflammatory activities of TA were regulated by the NF-kB and MAPKs signaling pathways (Fig 4). Our observation of the anti-IL-1β activity of TA in human osteoarthritis chondrocytes prompted us to further test the anti-osteoarthritis therapeutic effect in the MIA-induced OA rat model. As expected, the anti-IL-1β activity of TA not only relieved pain in the MIA-induced OA rat model, but also significantly lowered the OARSI/MANKIN score and decreased the expression of IL-1β in the tissue, showing a therapeutic effect on OA (Fig 5).

It is well known that TA has beneficial effects in relieving inflammatory pain, and TA creams have therapeutic effects in OA of the knee in clinical trials. Furthermore, many studies

have reported that TA and TA derivatives have therapeutic efficacy against OA in *in vivo* and *in vitro* OA models. Therefore, our results are consistent with those of previous reports on the suppression of OA-related factors by TA and various tannin derivatives [34–38]. Recently, the enthusiasm for IL-1 as a target in OA is rapidly dwindling due to *in vivo* models, showing a conflicting role for IL-1 molecule and some large double-blind randomized controlled clinical studies targeting IL-1 have failed [39]. Although the present study reports that IL-1β-direct targeting of TA showed anti-IL-1β activity by hindering the interaction between IL-1β and IL-1R1, further investigations are still needed to confirm whether TA can be applied as a therapeutic agent for other IL-1β-related diseases. In addition, the identification of the binding site of TA to IL-1β and IL-1R1, using molecular docking models, is also worthy of investigation. Nevertheless, this study is the first proves that TA can block the IL-1β–IL-1R1 interaction by binding to IL-1β and consequently inhibiting the activation of the IL-1β signaling pathway and IL-1β-related disease progression.

## Supporting information

**S1 Table. RT-qPCR primer sequences used in the present study.**
(DOC)

**S1 Raw images.**
(PDF)

## Author Contributions

**Conceptualization:** Mi-La Cho, Tae-Hwe Heo.

**Data curation:** Hae-Ri Lee, Young-Jin Jeong, Joong-Woon Lee, JooYeon Jhun, Hyun Sik Na, Keun-Hyung Cho, Mi-La Cho.

**Formal analysis:** Hae-Ri Lee, JooYeon Jhun, Hyun Sik Na, Keun-Hyung Cho.

**Funding acquisition:** Tae-Hwe Heo.

**Investigation:** Hae-Ri Lee, Young-Jin Jeong, Joong-Woon Lee, JooYeon Jhun, Hyun Sik Na, Keun-Hyung Cho.

**Methodology:** Hae-Ri Lee, JooYeon Jhun, Hyun Sik Na, Seok Jung Kim.

**Project administration:** Hae-Ri Lee, JooYeon Jhun, Seok Jung Kim, Mi-La Cho, Tae-Hwe Heo.

**Resources:** Hae-Ri Lee.

**Supervision:** Mi-La Cho, Tae-Hwe Heo.

**Validation:** Hae-Ri Lee, Young-Jin Jeong, JooYeon Jhun, Hyun Sik Na, Keun-Hyung Cho.

**Visualization:** Hae-Ri Lee, Young-Jin Jeong, JooYeon Jhun, Hyun Sik Na, Keun-Hyung Cho.

**Writing – original draft:** Hae-Ri Lee, Young-Jin Jeong.

**Writing – review & editing:** Hae-Ri Lee, Young-Jin Jeong, Tae-Hwe Heo.

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
