## [Decision Letter · Decision Letter 0]

8 Sep 2022

PONE-D-22-20982Tannic acid, an IL-1β-direct binding compound, ameliorates IL-1β-induced inflammation and cartilage degradation by hindering IL-1β-IL-1R1 interactionPLOS ONE

Dear Dr. Heo,

Thank you for submitting your manuscript to PLOS ONE. After careful consideration, we feel that it has merit but does not fully meet PLOS ONE’s publication criteria as it currently stands. Therefore, we invite you to submit a revised version of the manuscript that addresses the points raised during the review process.

We look forward to receiving your revised manuscript.

Kind regards,

Hassan Zmerly, MD PhD

Academic Editor

PLOS ONE

Journal Requirements:

2. To comply with PLOS ONE submissions requirements, please provide in the Methods section of your manuscript details regarding the method(s) of animal sacrifice used in this study.

"This work was supported by the Basic Science Research Program through the National Research Foundation of Korea (NRF) funded by the Ministry of Education, Science, and Technology (Grant number: NRF-2018R1A6A1A03025108 and 2021R1A2C2009782). This study was also supported by the Research Fund, 2020 of The Catholic University of Korea."

Reviewers' comments:

Reviewer's Responses to Questions

**Comments to the Author**

1. Is the manuscript technically sound, and do the data support the conclusions?

Reviewer #1: Yes

Reviewer #2: Yes

2. Has the statistical analysis been performed appropriately and rigorously? 

Reviewer #1: Yes

Reviewer #2: Yes

3. Have the authors made all data underlying the findings in their manuscript fully available?

Reviewer #1: Yes

Reviewer #2: Yes

4. Is the manuscript presented in an intelligible fashion and written in standard English?

Reviewer #1: Yes

Reviewer #2: Yes

5. Review Comments to the Author

Reviewer #1: Dear Authors,

Thank you for submitting the manuscript entitled "Tannic acid, an IL-1β-direct binding compound, ameliorates IL-1β-induced inflammation and cartilage degradation by hindering IL-1β-IL-1R1 interaction."

In this paper, authors investigated the role of tannic acid (TA) in the inhibition of the interaction between IL-1 β and IL-1R1, confirming the chondroprotective effects of TA due the suppression of MAPK and NF-κB signaling in IL-1β-induced OA rat model.

The paper has some strengths, but there are some critical points that need clarification and improvement especially concerning gene expression analyses.

Abstract

As reported in the submission guidelines, the abstract should not exceed 300 words. Please reduce its length trying to (i) describe the main objective of the study, (ii) explain how the study was done without methodological detail, and (iii) summarize the most important results and their significance.

Introduction

In the Introduction section (lines 85-86), authors assert that “many researchers have considered inhibiting the interaction between IL-1β and IL-1R1 as a therapeutic target for the treatment of various IL-1-related diseases.”

Please, list some examples of IL-1-related diseases, also adding appropriate recent references. For instance, since it is well-established that the most common side effects of CAR T cells therapy are cytokine release syndrome, developing new treatment options for CAR T cell-mediated toxicities based on IL-1 blockade has become an important part of immuno-oncological research.

Materials and Methods

Throughout the manuscript, please refer to the following reference for the correct nomenclature to use for human genes, especially for the use of italics (as also reported in the submission guidelines):

Bruford, E. A., Braschi, B., Denny, P., Jones, T., Seal, R. L., & Tweedie, S. (2020). Guidelines for human gene nomenclature. Nature genetics, 52(8), 754–758. https://doi.org/10.1038/s41588-020-0669-3.

2.7. Real-time polymerase chain reaction (RT-PCR)

This reviewer strongly recommends following the MIQE guidelines (The MIQE guidelines: minimum information for publication of quantitative real-time PCR experiments. Clin Chem. 2009 Apr;55(4):611-22. doi: 10.1373/clinchem.2008.112797), not only to use the correct nomenclature in the text, but also to enable other investigators to reproduce results. In this respect, please replace abbreviation qPCR with RT-qPCR if you performed reverse transcription–qPCR.

Moreover, although the use of a reference gene an internal control is the most common method for normalizing cellular mRNA data, the chosen reference gene should be stably expressed between treatment groups.

Hence, normalization against a single reference gene (e.g., GAPDH), as reported by the authors, is not acceptable unless the investigators present clear evidence that confirms its invariant expression under the experimental conditions described.

To this regard, has the M-value (the measure of expression stability) of your reference gene been calculated?

Have the authors purchased certified primers? In this case, please add information regarding the Company. On the other hand, if primers were custom designed, please add the length of amplicons in an appropriate Table.

2.12. Histopathological Analyses

Since histopathological analyses were performed on tissues of OA rat model, this reviewer suggests referring to the work of Gerwin et al. for a more appropriate use of the OARSI score: Gerwin, N.; Bendele, A.M.; Glasson, S.; Carlson, C.S. The OARSI histopathology initiative—Recommendations for histological assessments of osteoarthritis in the rat. Osteoarthr. Cartil. 2010, 18, S24–S34.

Results

3.1. Screening and affinity determination of IL-1β-blocking candidate

Line 270: what is the meaning of #397 after Fig. 1A within the brackets?

Concerning Figure 1A, please enhance the scale of the y-axis to show all the SD bars.

3.4. Effect of TA on the MAPK and NF-κB signaling pathways in IL-1β-stimulated human OA chondrocytes

Since TA showed a remarkable inhibitory effect on the IL-1β-induced phosphorylation of p38, ERK, JNK, p65, and IκBα, have authors tested specific inhibitors of MAPK and NF-κB signaling pathways, as negative controls?

Discussion

In the Discussion section, authors should comment on the fact which the enthusiasm for IL-1 as a target in OA is rapidly dwindling. In this regard, as reported by Prof. TL Vincent (doi: 10.12688/f1000research.18831.1), in vivo models show a conflicting role for IL-1 molecule; early studies using therapeutic approaches in large animal models show a benefit, but most murine studies fail to demonstrate protection where the ligands (IL-1α/β), the cytokine activator (IL-1–converting enzyme), or the receptor (IL-1R) have been knocked out. Recently, some large double-blind randomised controlled clinical studies targeting IL-1 have failed.

Reviewer #2: The manuscript is technically sound and well written. However, several points need be tackled, especially in the statistical analysis part and in the corresponding conclusions, in order to improve the manuscript. Additionally, some ambiguities exist and they need clarification. Below are my suggestions and concerns:

Introduction

1.Lines 101 – 105: the sentences are ambiguous. Can the authors please reformulate these sentences?

Materials & Methods

2.The authors are advised to check section 2.5 for possible errors in listing the performed ELISA assays. Did the author perform ELISA assay for MMP-3 & MMP13 or PGE2?

3.Section 2.1: Can the authors please define what did they use as positive and negative controls?

4.Lines 233-234: the sentence is ambiguous. Can the authors please reformulate this sentence?

Results

5.Authors are advised to check the abbreviations and signs used in the figures as many of these are not defined in the legends (For ex: +/-; NC; PC; NT…).

6.Figure 1A: The authors included only few of the 2303 tested molecules in the figure. Obviously, it would be impossible to include all the tested molecules, but why the authors chose to represent specifically the results of these compounds among others? And what are these compounds? As in the codes used for these compounds are not defined in the legend.

7.Lines 274-275: “the interaction between IL-1β and IL-1R1 was inhibited by TA in a dose dependent manner under harsh conditions”. The p-value presented in the figure 1B is for the comparison between the different TA concentrations and the negative control. If the authors are testing the dose-dependent effect, the p-values for the comparisons between the different concentrations of TA should be calculated and used to make conclusions.

8.Lines 277-279: the authors concluded that “TA reduced Il-1β induced SEAP secretion in a dose dependent manner” and that TA showed no cytotoxic effect at 100μΜ. It is not clear what the obtained p-values for these analyses are. And which comparison do they represent?

9.Figure 2, 3 & 4: Authors should identify which groups serve as positive and negative controls (in the materials and methods section).

10.Figure 3: It seems like the legend attributed to figure A corresponds to figure B and vice-versa. Can the authors please checks figure 3 legend?

11.Figure 4: Why did the authors studied the effect of TA on MAPK pathway with a single TA concentration of 2 μΜ while the TA effect on NF-κB pathway was studied with 1 & 2 μΜ of TA?

12.Figure 5: The authors compared the therapeutic effect of TA and vehicule as well as Celecobix and vehicule. It would be interesting to compare also the effect of TA to Celecobix to see if the effect of TA on the studied OA features is similar to Celecoxib or inferior/superior

13.Line 626: Can the author please check the number of rats in each group as in figure 5B it looks as there is 15 rats/group but in the text it is mentioned that there are 5 rats/group.

Discussion

14.Lines 376-378: the sentence is ambiguous. Can the authors please reformulate this sentence?

15.Lines 378-379: “We confirmed that the reduction in IL-1β signaling by TA was similar to that observed following treatment with soluble IL-1R1 or an anti-IL-1β neutralizing antibody”. However, in figures 2 & 3, the authors didn’t compare the effect of TA to the effect of soluble IL-1R1 or anti-IL-1β neutralizing antibody. The p-values represented in the figures are for the comparison between different concentrations of TA & IL-1β treated group.

16.Lines 388 -389: lack references.

6. PLOS authors have the option to publish the peer review history of their article (what does this mean?). If published, this will include your full peer review and any attached files.

Reviewer #1: No

Reviewer #2: No

---

## [Author Response · Author response to Decision Letter 0]

18 Oct 2022

PLoS One

Responses to the reviewer’s comments:

We are grateful to the reviewers for their valuable feedback and insightful comments that helped us to improve the quality of our manuscript. We have carefully considered all the comments and revised the manuscript accordingly. Our point-wise responses to the reviewers’ comments are given below. We hope that the response meets the reviewers’ expectations. We are ready to continue working with the editor and reviewers to bring this paper closer to publication.

Our responses are detailed below: 

Reviewer 1:

 Abstract

As reported in the submission guidelines, the abstract should not exceed 300 words. Please reduce its length trying to (i) describe the main objective of the study, (ii) explain how the study was done without methodological detail, and (iii) summarize the most important results and their significance.

Response: Thank you for the comments. We have revised the abstract accordingly. The new version of abstract is included in the revised manuscript (Revised lanes 37-60). 

Introduction

In the Introduction section (lines 85-86), authors assert that “many researchers have considered inhibiting the interaction between IL-1β and IL-1R1 as a therapeutic target for the treatment of various IL-1-related diseases.”

Please, list some examples of IL-1-related diseases, also adding appropriate recent references. For instance, since it is well-established that the most common side effects of CAR T cells therapy are cytokine release syndrome, developing new treatment options for CAR T cell-mediated toxicities based on IL-1 blockade has become an important part of immuno-oncological research.

Response: We completely agree with your suggestion. We have included this information in the revised Introduction section. (Revised lines 83-88):” For instance, anakinra bind to IL-1R1 and inhibit its interaction with IL-1β. The anti-IL-1β activity of anakinra was shown to reduce mortality via anti-hyperinflammatory efficacy in patients with severe COVID-19 pneumonia [4]. Anakinra also showed anti-hyperinflammatory efficacy in other hyperinflammatory situations, including the cytokine release syndrome and immune effector cell-associated neurotoxicity syndrome observed during antitumoral chimeric antigen receptor (CAR) T-cell therapy [5, 6].”

Materials and Methods

Throughout the manuscript, please refer to the following reference for the correct nomenclature to use for human genes, especially for the use of italics (as also reported in the submission guidelines):

Bruford, E. A., Braschi, B., Denny, P., Jones, T., Seal, R. L., & Tweedie, S. (2020). Guidelines for human gene nomenclature. Nature genetics, 52(8), 754–758. https://doi.org/10.1038/s41588-020-0669-3

Response: Thank you for your comments. We have corrected the nomenclature throughout the entire manuscript.

2.7. Real-time polymerase chain reaction (RT-PCR)

This reviewer strongly recommends following the MIQE guidelines (The MIQE guidelines: minimum information for publication of quantitative real-time PCR experiments. Clin Chem. 2009 Apr;55(4):611-22. doi: 10.1373/clinchem.2008.112797), not only to use the correct nomenclature in the text, but also to enable other investigators to reproduce results. In this respect, please replace abbreviation qPCR with RT-qPCR if you performed reverse transcription–qPCR.

Moreover, although the use of a reference gene an internal control is the most common method for normalizing cellular mRNA data, the chosen reference gene should be stably expressed between treatment groups.

Hence, normalization against a single reference gene (e.g., GAPDH), as reported by the authors, is not acceptable unless the investigators present clear evidence that confirms its invariant expression under the experimental conditions described.

To this regard, has the M-value (the measure of expression stability) of your reference gene been calculated?

Response: Thank you for your valuable comment. We completely agree with your suggestion. Although we did not further evaluate the M-value of GAPDH as a reference gene, it was reported that GAPDH showed stable gene expression in IL-1β-treated human chondrocytes [1]. Thus, GAPDH was chosen as the reference housekeeping gene based on the majority of previous studies on chondrocyte gene expression. 

In addition, using two other reference genes, RPL13a and YWHAZ [2], which are the most stable expressed genes for reference in human chondrocytes, we tried to normalize some of the tested gene expressions in Figure 2A and 2B in the manuscript. As shown below, the normalization results using two additional reference genes, RPL13a(A) and YWHAZ(B), were consistent with the existing normalization results using GAPDH as a reference gene. 

[1] Toegel S et. al. Selection of reliable reference genes for qPCR studies on chondroprotective action. BMC Mol Biol. 2007; 8:13. DOI: 10.1186/1471-2199-8-13.

[2] Akira Ito et. al. Evaluation of reference genes for human chondrocytes cultured in several different thermal environments. Int J Hyperthermia. 2014;30(3):210-6. DOI:10.3109/02656736.2014.906048.

Have the authors purchased certified primers? In this case, please add information regarding the Company. On the other hand, if primers were custom designed, please add the length of amplicons in an appropriate Table.

Response: Thank you for the comment. We have added the information of primer sequences in Table S1. Please see methods section and supplementary information file.

2.12. Histopathological Analyses

Since histopathological analyses were performed on tissues of OA rat model, this reviewer suggests referring to the work of Gerwin et al. for a more appropriate use of the OARSI score: Gerwin, N.; Bendele, A.M.; Glasson, S.; Carlson, C.S. The OARSI histopathology initiative—Recommendations for histological assessments of osteoarthritis in the rat. Osteoarthr. Cartil. 2010, 18, S24–S34.

Response: We appreciate your valuable comments. However, this reference is suitable for the DMM or ACLT surgery model. In case of the MIA-induced osteoarthritis model, it shows severe bone destruction and cartilage damage. Therefore, the chemically induced model requires a scoring analysis focusing on cell death, bone destruction, and cartilage damage (Osteoarthritis and Cartilage (2006) 14, 13e29 Table III).

Results

3.1. Screening and affinity determination of IL-1β-blocking candidate

Line 270: what is the meaning of #397 after Fig. 1A within the brackets?

Response: We apologize for the error. We replaced the labels of Figure 1A #397 to Tannic acid. Please see revised manuscript and Fig. 1A.

Concerning Figure 1A, please enhance the scale of the y-axis to show all the SD bars. 

Response: We have replaced the Figure 1A to improve clarity. 

3.4. Effect of TA on the MAPK and NF-κB signaling pathways in IL-1β-stimulated human OA chondrocytes

Since TA showed a remarkable inhibitory effect on the IL-1β-induced phosphorylation of p38, ERK, JNK, p65, and IκBα, have authors tested specific inhibitors of MAPK and NF-κB signaling pathways, as negative controls?

Response: Thank you for your reasonable comments. Although we did not include negative control related data in previous manuscript, we have also compared the effects of TA with specific inhibitors of MAPK and NF-kB. IL-1β-induced phosphorylations of p38, ERK, JNK, p65, and IkBα were inhibited by TA as shown in Figures 4A and 4B; however, its inhibitory effects were lower than inhibition by MAPK or NF-kB specific inhibitors (SB203580; p38 inhibitor, U0126; ERK inhibitor, SP600125; JNK inhibitor, and Bay11-7082; NF-kB inhibitor).

Discussion

In the Discussion section, authors should comment on the fact which the enthusiasm for IL-1 as a target in OA is rapidly dwindling. In this regard, as reported by Prof. TL Vincent (doi: 10.12688/f1000research.18831.1), in vivo models show a conflicting role for IL-1 molecule; early studies using therapeutic approaches in large animal models show a benefit, but most murine studies fail to demonstrate protection where the ligands (IL-1α/β), the cytokine activator (IL-1–converting enzyme), or the receptor (IL-1R) have been knocked out. Recently, some large double-blind randomised controlled clinical studies targeting IL-1 have failed.

Response: We agree with your suggestion. We have included this information in Discussion (Revised lines 398-400). “Recently, the enthusiasm for IL-1 as a target in OA is rapidly dwindling due to in vivo models, showing a conflicting role for IL-1 molecule and some large double-blind randomized controlled clinical studies targeting IL-1 have failed [39].”

Reviewer #2: 

Introduction

1.Lines 101 – 105: the sentences are ambiguous. Can the authors please reformulate these sentences?

Response: We have rewritten these sentences in the revised manuscript (Revised lines 103-107). “Since accumulating evidence has demonstrated that IL-1β-mediated inflammation is important in the progression of OA, blocking IL-1β signaling may be used as an additional therapeutic strategy in drugs for OA. Therefore, it would be a good strategy to use IL-1β-mediated OA progression as an in vivo research model to evaluate the therapeutic efficacy of compounds with anti-IL-1β activity.”

Materials & Methods

2.The authors are advised to check section 2.5 for possible errors in listing the performed ELISA assays. Did the author perform ELISA assay for MMP-3 & MMP13 or PGE2?

Response: We have corrected relevant information regarding ELISA assay in the revised manuscript section 2.5.

3.Section 2.1: Can the authors please define what did they use as positive and negative controls?

Response: Thank you for your comment. We have defined the positive and negative control in the revised manuscript.

4.Lines 233-234: the sentence is ambiguous. Can the authors please reformulate this sentence?

Response: We have rewritten the sentence. Please see revised manuscript (Revised lines 222-223). “The weights on the unguided and guided legs were determined. The percentage weight balance was obtained by comparing legs with and without OA.”

Results

5.Authors are advised to check the abbreviations and signs used in the figures as many of these are not defined in the legends (For ex: +/-; NC; PC; NT…).

Response: Thank you for the comments. We have corrected the abbreviations and signs in the revised figure legend (Revised lines 562-565, 592, 603, and 611). “Binding of IL-1β and IL-1R1 without any single compound was used as a positive control (P.C.), and nonspecific binding of IL-1β and HRP-conjugated anti-human IgG secondary antibody without any single compound and IL-1R1 was used as a negative control (N.C.). ** p <0.01 compared to the positive control group.” and ““+” indicates treated and “-” indicates un-treated.”

6.Figure 1A: The authors included only few of the 2303 tested molecules in the figure. Obviously, it would be impossible to include all the tested molecules, but why the authors chose to represent specifically the results of these compounds among others? And what are these compounds? As in the codes used for these compounds are not defined in the legend.

Response: Since we screened 2303 natural compound libraries, it would be impossible to include all the results of tested molecules. Therefore, we selected the anterior and posterior compound portions, including TA, from among the compound results within the library, and presented the results in Figure 1A. As suggested by the reviewer’s comment, we added the information of additional tested compounds in Figure legend (Revised lines 565-567). “Among 2303 natural compound, only some compounds were used for screening results, and the numbers 316 to 443 mean serial numbers represent the natural compounds in the TargetMol L-6000 library.” 

7.Lines 274-275: “the interaction between IL-1β and IL-1R1 was inhibited by TA in a dose dependent manner under harsh conditions”. The p-value presented in the figure 1B is for the comparison between the different TA concentrations and the negative control. If the authors are testing the dose-dependent effect, the p-values for the comparisons between the different concentrations of TA should be calculated and used to make conclusions.

Response: Thank you for pointing this out. As suggested by the reviewer’s comment, comparisons between the different TA concentrations were performed against each other and the new information was included in the revised Figure 1B legend (Revised line 570-571). “## p <0.01 compared to the positive control group. ** p <0.01 compared to treatment groups with different TA concentrations (10 or 40 μM).”

8.Lines 277-279: the authors concluded that “TA reduced IL-1β induced SEAP secretion in a dose dependent manner” and that TA showed no cytotoxic effect at 100 μΜ. It is not clear what the obtained p-values for these analyses are. And which comparison do they represent?

Response: The comparisons between the different TA concentrations were performed against each other and we have added new information in the revised manuscript (Revised lines 258-261), Figure 1C and Figure legend 1. “TA reduced IL-1β-induced SEAP secretion (Fig. 1C left panel, IC50 = 13.31 µM) and anti-IL-1β activity of TA showed a dose-dependent manner (Fig. 1C, right panel). In addition, the cytotoxic effect of TA did not show a significant difference up to 100 μM compared to the medium alone group (Fig. 1D).”

9.Figure 2, 3 & 4: Authors should identify which groups serve as positive and negative controls (in the materials and methods section).

Response: We have defined positive and negative control in the revised Material and Methods section (Revised line 175-176). “Recombinant human IL-1R1 Fc or anti-IL-1β neutralizing antibody were used as a positive control and medium only group was used as a negative control.”

10.Figure 3: It seems like the legend attributed to figure A corresponds to figure B and vice-versa. Can the authors please checks figure 3 legend?

Response: We have corrected legend of Figure 3 in the revised manuscript (Revised line 598-602). “(A) mRNA expressions of MMPs, ADAMTSs, COL2A1, and ACAN were detected using RT-qPCR and the relative quantity of each gene expression was normalized to the internal reference gene GAPDH. (B) Protein expressions of MMPs, ADAMTSs, COL2A1, and ACAN were detected using western blot analysis and β-actin was used as a loading control. # p < 0.05 and ## p < 0.01 compared with medium only control group.” 

11.Figure 4: Why did the authors studied the effect of TA on MAPK pathway with a single TA concentration of 2 μΜ while the TA effect on NF-κB pathway was studied with 1 & 2 μΜ of TA?

Response: Since the NF-κB pathway plays an important role in the pathogenesis of OA, the TA effect experiment on the NF-κB pathway was first conducted according to the TA concentration. Consequently, 2 μM of TA showed a more dramatic inhibitory efficacy, thus the MAPK pathway proceeded only with a single TA concentration of 2 μΜ. 

We have modified the results for the NF-κB pathway to match MAPK pathway tested with a single concentration of 2 μM of TA to improve clarity.

12.Figure 5: The authors compared the therapeutic effect of TA and vehicle as well as Celecoxib and vehicle. It would be interesting to compare also the effect of TA to Celecoxib to see if the effect of TA on the studied OA features is similar to Celecoxib or inferior/superior 

Response: Comparisons of the therapeutic effects between the TA and Celecoxib-treated group were performed. The therapeutic efficacy of TA showed a similarity to the celecoxib-treated positive control, and the p-value between the TA and Celecoxib-treated group was not significant. 

13.Line 626: Can the author please check the number of rats in each group as in figure 5B it looks as there is 15 rats/group but in the text it is mentioned that there are 5 rats/group.

Response: To objectively evaluate histological scoring, each of the three evaluators who did not know the experimental group performed histological analysis, so the meaning of 15 dots refers to the results of each of the three evaluators for 5 rat samples per group. To avoid confusion, the results of the three evaluators were averaged and corrected for bar graphs. And we replaced the Fig. 5B. Please see revised Fig. 5B.

Discussion

14.Lines 376-378: the sentence is ambiguous. Can the authors please reformulate this sentence?

Response: We have rewritten the sentence. Please see revised manuscript (Revised lines 358-360). “TA showed an anti-IL-1β bioactivity against expression of catabolic and anabolic factors in IL-1β-stimulated human OA chondrocytes via inhibition of IL-1β-induced MAPK and NF-κB activation.”

15.Lines 378-379: “We confirmed that the reduction in IL-1β signaling by TA was similar to that observed following treatment with soluble IL-1R1 or an anti-IL-1β neutralizing antibody”. However, in figures 2 & 3, the authors didn’t compare the effect of TA to the effect of soluble IL-1R1 or anti-IL-1β neutralizing antibody. The p-values represented in the figures are for the comparison between different concentrations of TA & IL-1β treated group.

Response: Thank you for your pointing this out. As suggested by the reviewer’s comment, we have corrected Discussion section in the revised manuscript.

16.Lines 388 -389: lack references.

Response: We have added the reference information in the revised manuscript.

---

## [Decision Letter · Decision Letter 1]

2 Feb 2023

Tannic acid, an IL-1β-direct binding compound, ameliorates IL-1β-induced inflammation and cartilage degradation by hindering IL-1β-IL-1R1 interaction

PONE-D-22-20982R1

Dear Dr. Tae-Hwe Heo,

We’re pleased to inform you that your manuscript has been judged scientifically suitable for publication and will be formally accepted for publication once it meets all outstanding technical requirements.

Kind regards,

Hassan Zmerly, MD PhD

Academic Editor

PLOS ONE

Additional Editor Comments (optional):

Reviewers' comments:

Reviewer's Responses to Questions

**Comments to the Author**

1. If the authors have adequately addressed your comments raised in a previous round of review and you feel that this manuscript is now acceptable for publication, you may indicate that here to bypass the “Comments to the Author” section, enter your conflict of interest statement in the “Confidential to Editor” section, and submit your "Accept" recommendation.

Reviewer #1: All comments have been addressed

2. Is the manuscript technically sound, and do the data support the conclusions?

Reviewer #1: Yes

3. Has the statistical analysis been performed appropriately and rigorously? 

Reviewer #1: Yes

4. Have the authors made all data underlying the findings in their manuscript fully available?

Reviewer #1: Yes

5. Is the manuscript presented in an intelligible fashion and written in standard English?

Reviewer #1: Yes

6. Review Comments to the Author

Reviewer #1: The authors have addressed the issues raised previously, and the manuscript is suitable for publication in its current form.

7. PLOS authors have the option to publish the peer review history of their article (what does this mean?). If published, this will include your full peer review and any attached files.

Reviewer #1: No

---

## [Editor Report · Acceptance letter]

11 Apr 2023

PONE-D-22-20982R1 

Tannic acid, an IL-1β-direct binding compound, ameliorates IL-1β-induced inflammation and cartilage degradation by hindering IL-1β-IL-1R1 interaction 

Dear Dr. Heo:

I'm pleased to inform you that your manuscript has been deemed suitable for publication in PLOS ONE. Congratulations! Your manuscript is now with our production department. 

Kind regards, 

on behalf of

Professor Hassan Zmerly 

Academic Editor

PLOS ONE